# A Symbolic Framework for Evaluating Mathematical Reasoning with Transformers

## Abstract

This paper proposes a methodology for generating synthetic mathematical derivations via a computer algebra system to evaluate the generalisability of Transformers in symbolic and quantitative reasoning problems, and provides a general framework for building large-scale and high-quality benchmarks in the mathematical domain. In the context of classification tasks involving multi-step annotated derivations (spanning 18 mathematical operators), we leverage the framework to compare the mathematical capabilities of GPT-4, GPT-3.5, and a canon of fine-tuned BERT models, and explore the relationship between specific operators and generalisation failure. Surprisingly, the average in-distribution performance of BERT models surpasses GPT-3.5, and rivals GPT-4, yet simple symbolic perturbations reduce BERT scores by up to 80 F1 points. The results suggest that the in-distribution performance and generalisability of smaller open-source models may potentially rival GPT in narrow mathematical domains by incorporating appropriately structured discourse-level relations during training, and highlight a shared weakness between BERT and GPT involving a relative inability to decode dependency relations involving indirect references to mathematical entities. We release the data generation framework along with all the resulting datasets and fine-tuned models[1].

## 1 Introduction

Out-of-distribution (OOD) generalisation in Transformers (Vaswani et al., 2017) is a fundamental property for domain-specific/specialised natural language inference (Schlegel et al., 2023; Belinkov, 2022; Teney et al., 2020) in areas which require rigorous and controlled reasoning such as mathematics, physics, biomedicine, and software verification (Frieder et al., 2023; Lee et al., 2022; Valentino et al., 2022b; Lewkowycz et al., 2022; Drori et al., 2022; Welleck et al., 2021; Kumar et al., 2020). Various strategies have been proposed to evaluate model generalisability, including direct input manipulation (Rozanova et al., 2023b; Stolfo et al., 2022; Nie et al., 2020; Kaushik et al., 2019) and probing on the internal representation (Rozanova et al., 2023a; Ravichander et al., 2021; Elazar et al., 2021; Veitch et al., 2020). This paper considers input interventions through syntactic and semantic perturbations to mathematical text. Current interventional approaches are challenged by the difficulty of isolating confounding factors, and formalising the expected causal mechanisms that underpin the models' predictions (Rozanova et al., 2023b; Stolfo et al., 2022; Ribeiro et al., 2020; Kaushik et al., 2019). Particularly in the mathematical domain, these hurdles impact the scope and reliability of causality and robustness studies (Pearl, 2009; Shreya et al., 2022).

To tackle existing limitations, we leverage the rich environment of symbolic engines to design a data generation and evaluation framework that generates mathematical reasoning steps possessing diverse symbolic properties and produces equation derivations at scale. Strict symbolic rules offer a systematic approach to perturbing mathematical reasoning and hence evaluating the OOD generalisation of neural models in various tasks. This allows us to explore deep relationships between semantic and syntactic elements of math reasoning and model generalisability across diverse subdomains, extending beyond the limited interventional scope of previous works (Stolfo et al., 2022; Welleck et al., 2022; Patel et al., 2021; Ribeiro et al., 2020; Kaushik et al., 2019; Yao et al., 2021). In this work we explore generalisability in the context of multi-hop equational reasoning and sequence classification tasks, where sequences of mathematical operators are applied to premises and prior equations to advance derivations, and provide model input.

---

[1] https://github.com/anonymous/TBA

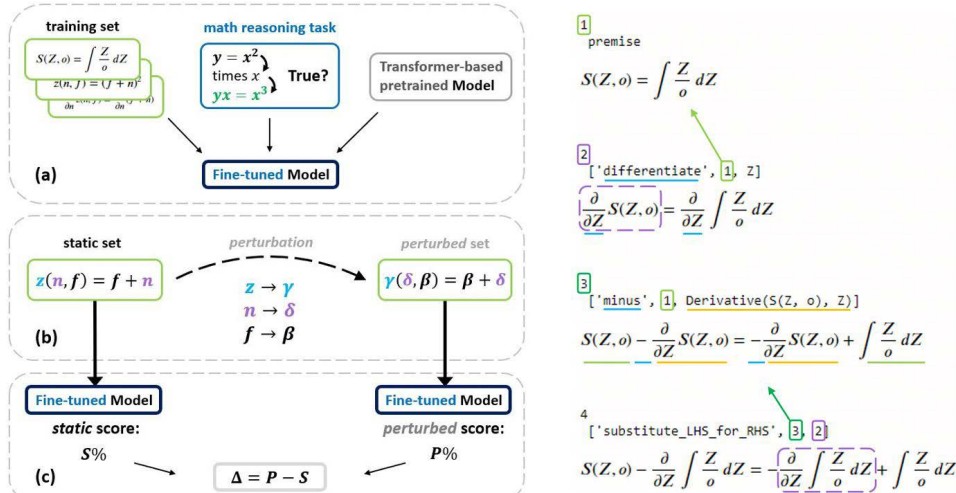

Figure 1: An overview of the proposed framework (left) and example of a generated derivation (right). We leverage computer algebra to generate large-scale training data for mathematical reasoning tasks **(a)** and apply systematic perturbations to examples from a static test set to form a perturbed test set **(b)**. The static evaluation scores are compared with scores on the perturbed set, given some metric, to determine model robustness and generalisation **(c)**. In the derivation (right), colours highlight long-range dependencies.

Additionally, we dialogue with an impending data scarcity problem, where high-quality data is forecast to be outpaced by the training needs of models within the decade (Villalobos et al., 2022). Symbolic engines facilitate the generation of annotated mathematical reasoning, which allows the construction of high-quality datasets for various tasks. We combine (18) symbolic operators with hand-crafted rules that guide the exploration of equational state spaces and generate derivations, then perturb and adapt them for exemplar entailment tasks. In this case, these are sequence classification tasks that focus on operator usage in reasoning chains.

To demonstrate our approach, we fine-tune a canon of BERT-based models used in mathematical language processing (Li et al., 2023; McNichols et al., 2023; Zhong et al., 2022; Meadows & Freitas, 2022), and few-shot prompt GPT-3.5 and GPT-4, to determine their capacity for recognising coherent math reasoning, and to abstract fundamental properties impacting their ability to generalise. To summarise, the paper offers the following contributions:

**1.** An approach to generating annotated derivations of controllable complexity levels, involving premise equation generation (Algorithm 1) and the sequential application of operators to prior equations to derive new results (Algorithm 2).

**2.** A systematic and scalable methodology to perturb various aspects of mathematical data including syntax and semantics. We outline a number of simple perturbations in this initial case.

**3.** An experimental framework for training models on mathematical reasoning tasks and evaluating their robustness, including dataset generation, systematic perturbation, training, and evaluation (Fig. 1).

**4.** Example instantiation of the framework involving sequence classification tasks. The generated datasets include static and perturbed derivations totalling over 200K examples.

**5.** An extensive comparative evaluation of various BERT-based and GPT models culminating in a discussion relating the limited generalisability of models with respect to key operators and mathematical content.

To the best of our knowledge, this work is the first to propose a general symbolic engine-based framework for producing large-scale and highly controllable benchmarks for multi-step mathematical reasoning (in both LaTeX and SymPy notation).

## 2 RELATED WORK

**Computer algebra.** SymPy (Meurer et al., 2017) is a computer algebra system used in conjunction with a number of language processing methods. For example, Chen et al. (2022) solve numerical reasoning tasks including simple math elements such as numbers, by chain-of-thought prompting language models to generate SymPy solvable code. Mandlecha et al. (2022) use SymPy to generate data for answering questions ranging from arithmetic to calculus without exploring generalisability aspects. Hu & Yu (2022) solve a similar array of problems from a large-scale dataset (Saxton et al., 2019), and test for generalisability to an extrapolation set of problems. Drori et al. (2022) fine-tune the decoder model, Codex (Chen et al., 2021), on a dataset of questions from MIT's university-level mathematics courses, generating SymPy solution code. Lample & Charton (2019) train a model to integrate and solve differential equations more successfully than computer algebra systems, but as noted elsewhere (Davis, 2019) they do not explore OOD performance. Welleck et al. (2022) conduct similar experiments using a single model and a single operator (integration) on a single task. We consider 18 operations, 7 models, multiple tasks, and emphasize perturbations applied to multi-step reasoning.

**Reasoning with mathematical language.** Transformers (Saxton et al., 2019; Clark et al., 2020; Rabe et al., 2020) defined the state-of-the-art (SotA) in multiple subdomains and tasks in mathematical language processing (Meadows & Freitas, 2022; Lewkowycz et al., 2022; Drori et al., 2022). Transformer encoder models obtain SotA performance in variable typing (Ferreira et al., 2022; Lai et al., 2022), formula search (Zhong et al., 2022; Peng et al., 2021), natural language premise selection (Valentino et al., 2022a; Tran et al., 2022), and retrieval-based math question answering (Reusch et al., 2022; Novotný & Štefánik, 2022), among other tasks. The evaluation of the mathematical capabilities of GPT models, as well as the comparison between GPT and smaller fine-tuned models when deriving equations, has been considered elsewhere (Meadows et al., 2023; Frieder et al., 2023).

**Data augmentation and evaluation frameworks.** Many approaches exist related to evaluating the mathematical and symbolic capabilities and robustness of models. Stolfo et al. (2022) perturb elements of math word problems (Liang et al., 2022) such as numerical operands of implicit arithmetic operations, and natural language, inspired by related work (Pearl, 2022; Christiansen et al., 2021; Patel et al., 2021; Ribeiro et al., 2020) in causal analysis. Similar to other work Welleck et al. (2022), their approach focuses on one or two task-dependent perturbations. Our approach to generating and perturbing data is largely task-independent, and allows for the complex augmentation of operators, variables, expressions, and equations in multi-hop reasoning chains.

## 3 GENERATING AND PERTURBING DERIVATIONS WITH SYMBOLIC ALGEBRA ENGINES

In this section, we describe the methodology for generating synthetic mathematical derivations from a vocabulary of symbols and a set of operators. The set of operators includes addition, subtraction, multiplication, division, exponentiation, $\cos$, $\sin$, $\log$, $\exp$, operations for setting up derivatives and integrals, expression substitutions, and operations for defining premises. An example derivation is given in Fig. 1.

### 3.1 PREMISE GENERATION

A derivation represents a sequence of operations initially applied to premise equations, as shown in Fig. 1. To generate premises we adopt a vocabulary and a set of operators defined within the symbolic engine. The vocabulary includes uppercase and lowercase English characters, excluding {i, e, d, O} to avoid overlap with standard mathematical notation. Operators are separated by their arity. For example, the symbols $Z$ and $o$ are sampled from the vocabulary and used as operands for the 2-arity operator "divide". Then, $Z$ is sampled from the vocabulary as an operand for the 1-arity operator "integrate". This expression becomes $\int \frac{Z}{o} dZ$, and consists of the free symbols $Z$ and $o$. This is the RHS of the premise equation. To form the LHS, a function symbol is sampled from the vocabulary, in this case $S$, and the two free symbols are assigned as variables. The LHS and RHS are themselves inputted as arguments of an equation operation, and the premise (1) is obtained. A formal description of the premise generation process is given by Algorithm 1 in Appendix D.

## 3.2 DERIVATION GENERATION

To summarise the primary mechanism for the derivation generation approach, operators are classified by their arity $\in [0, 2]$ which determines step annotations. A sampled operator is then applied to expressions to each side of a sampled equation to generate new equations.

For example, starting from the premise in Fig. 1 (right), given by $(Z, o) = \int \frac{Z}{o} dZ$, the 2-arity class can be selected and the operation "differentiate" can be chosen from the list of operators matching that arity. Subsequently, the algorithm randomly selects a variable to which to apply the operation (i.e., $Z$). The generated annotation ['differentiate', 1, Z], therefore, means that the operator "differentiate" was applied to operand equation (1), with respect to $Z$ to yield $\frac{\partial}{\partial Z} S(Z, o) = \frac{\partial}{\partial Z} \int \frac{Z}{o} dZ$. Similarly, the notation ['minus', 1, Derivative(S(Z,o), Z)] means that the 2-arity operation "minus" was selected, the operator and the LHS of (2) is selected as the second operand. This step-wise procedure repeats up to a predefined number of steps to produce a full derivation with structured inter-statement relations and references, where the correctness of step calculations are guaranteed by the computer algebra engine. We describe this formally in Algorithm 2 with a more detailed description of hyperparameters and equation sampling in Appendix E.

## 3.3 PERTURBATIONS

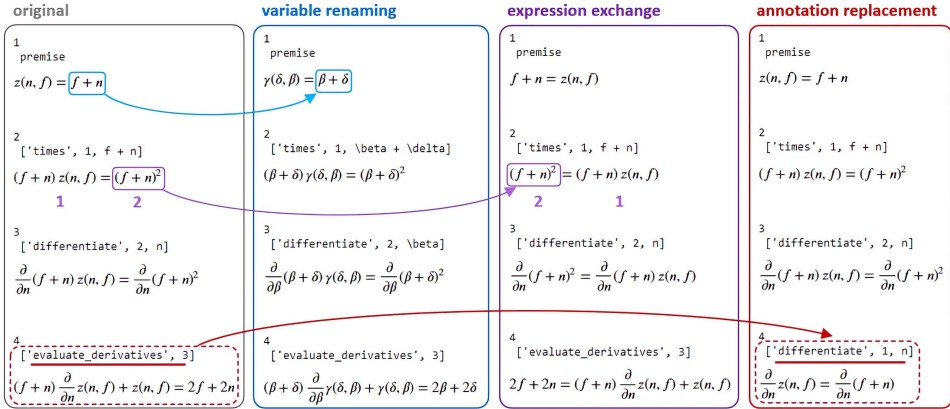

Figure 2: Example perturbations applied to a derivation using computer algebra.

To perturb LaTeX sequences, the examples in the static set are *re-interpreted* by the computer algebra engine using SymPy's `srepr` tree representation[2]. The one-to-one mapping between LaTeX and Sympy allows for derivations to be perturbed with respect to a target property of interest, and represented using different formats. In this paper, we consider four different perturbations for evaluation (Fig. 2). However, the compatibility with the computer algebra system facilitates perturbed reasoning that ranges from small-scale interventions to single variables through to long-range interventions targeting complex semantic relationships between any number of distant sequence elements. For instance, one may choose to only perturb reasoning chains that involve a premise renaming operation followed directly by integration, or square a variable and propagate that change through the entire reasoning chain. The perturbations adopted in our evaluation are as follows:

**Variable Renaming (VR)**. For each example in the static set, we uniquely map each symbol to an out-of-vocabulary symbol sampled from 10 Greek letters (*e.g.*, $E(n, x) = n + x$ becomes $\alpha(\beta, \gamma) = \beta + \gamma$).

**Expression Exchange (EE)**. For each example in the static set, we swap expressions either side of the equality (*e.g.*, $E(n, x) = n + x$ becomes $n + x = E(n, x)$). This reverses the overrepresentation of LHS functions in the static set.

**Annotation Replacement (AR)**. Each example in the static set contains a positive and negative final equation. For each example, the operator and operands (and hence the annotation) responsible for generating the negative equation are calculated, replacing the corresponding annotation in the sequence and swapping the label (i.e. from positive to negative and vice-versa).

---

[2]https://docs.sympy.org/latest/tutorials/intro-tutorial/manipulation.html

**Equation Conversion (EC).** If a sequence consists of a chain such as $\log(x)$ [SEP] $x$ [SEP] $\frac{1}{x}$, and the implicit operation is differentiation (*e.g.*, Fig. 3(b)), a random symbol is sampled from the vocabulary (*e.g.*, $Q$), and the sequence becomes $Q(x) = \log(x)$ [SEP] $x$ [SEP] $\frac{dQ(x)}{dx} = \frac{1}{x}$. If integrating, then the (negative) sequence becomes $\frac{dQ(x)}{dx} = \log(x)$ [SEP] $x$ [SEP] $Q(x) = \frac{1}{x}$.

# 4 TASKS

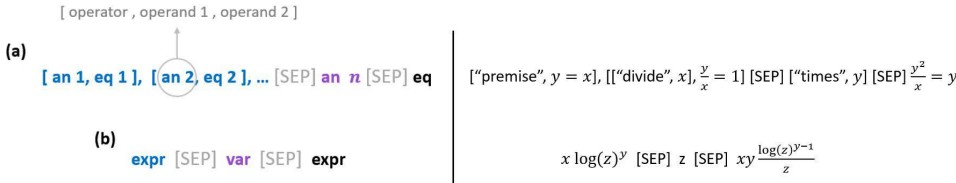

Figure 3: Input formats for the two sequence classification tasks (left): *Derivation Step Classification* (a) and *Calculus Classification* (b). Specific inputs are shown on the right.

We instantiate the general framework described in Section 3 in the context of two sequence classification tasks, where models must predict whether the final expression or equation in the sequence follows from the prior context. The main data generation algorithm outputs a derivation (Alg. 2) in LaTeX and SymPy (Fig. 1) which must then be adapted for specific tasks. Training and prompt exploration details for each task are described in Appendix A. Task dataset sizes and sequence construction details are described in Appendix B.

**Derivation Step Classification.** Fig. 3(a) describes the model input format for this task. The aim of the task is to predict the final result of a sequence of operations. Each individual step consists of an equation and an annotation that describes the details of the applied operation (Fig. 1). Negative examples are generated by applying either a different operation, or the same operation with different operands. Therefore, to solve this task while being robust to perturbations, a model must learn the necessary equation dependencies required to form the final equation in the derivation, guided by the final annotation. In experiments we consider derivations composed of up to four steps.

**Calculus Classification.** Fig. 3(b) describes the model input for this task, which consists in a single-step calculation of derivatives and integrals (related to Lample & Charton (2019); Welleck et al. (2022)). Separately to the previous task, here we aim to evaluate the capability of the models to perform a single inference step without access to the operation annotation. Therefore, to build the dataset, we generate a premise expression containing *at least two* variables, and use as the ground truth the resulting expression after differentiating or integrating with respect to a randomly selected variable. The negative examples are generated by sampling from a list of alternative premises that include the result of differentiating/integrating premises either by fixing the variable and changing the expression, or vice versa.

**Generalisation to simpler mathematics.** A model that can sufficiently generalise the mathematical rules underlying the tasks should be able to solve (on average) mathematically less complex versions of problems encountered during training. To this end, in *Derivation Step Classification*, we evaluate models exposed to derivations with a fixed step count on a set of derivations composed of a lower number of steps. This is represented in the **s - 1** and **s - 2** columns in Tab. 1 given initial step count, **s**. In *Calculus Classification*, where models are exposed to examples comprising *at least two* variables, (*e.g.*, $\cos(ax) - z$) we generate a set of easier problems with 1.5k examples that consist of only one variable (*e.g.*, $\cos(x)$).

# 5 EVALUATION

In this section, we present and discuss the various scores obtains by range of fine-tuned and pre-trained Transformer-based models on the classification tasks in Tables 1 and 2. Additional details regarding experimental setup and reproducibility are discussed in Appendix A.

**GPT-4 rivals in-distribution performance of fine-tuned BERT-based models while demonstrating better generalisation.** Assuming a suitably descriptive few-shot prompt, where necessary context is provided through either the task description or in-context examples (Appendix A), GPT-4 can rival

| | Static | | VR | | EE | | AR | | s - 1 | | s - 2 | |
|---|---|---|---|---|---|---|---|---|---|---|---|---|
| | Acc | F1 | Acc | F1 | Acc | F1 | Acc | F1 | Acc | F1 | Acc | F1 |
| BERT-base-uncased (**s=2**) | 87.7 | 88.9 | 87.0 | 88.1 | 87.0 | 88.0 | 87.5 | 88.7 | - | - | - | - |
| BERT-base-uncased (**s=3**) | 78.9 | 78.7 | 71.9 | 71.0 | 69.1 | 66.0 | 53.7 | 50.6 | 68.4 | 69.0 | - | - |
| BERT-base-uncased (**s=4**) | 58.8 | 63.6 | 55.0 | 60.3 | 56.4 | 60.3 | 42.4 | 48.1 | 65.7 | 62.2 | 52.8 | 29.8 |
| BERT-base-cased (**s=2**) | 87.2 | 88.5 | 81.9 | 83.2 | 85.3 | 86.1 | 85.5 | 87.2 | - | - | - | - |
| BERT-base-cased (**s=3**) | 78.2 | 77.3 | 68.8 | 64.5 | 65.0 | 58.9 | 54.5 | 49.6 | 54.6 | 30.5 | - | - |
| BERT-base-cased (**s=4**) | 66.8 | 71.7 | 58.5 | 61.5 | 62.6 | _67.2_ | 43.3 | 53.1 | 71.9 | 73.9 | 54.3 | 21.8 |
| MathBERT (**s=2**) | 83.2 | 82.0 | 76.2 | 70.6 | 79.0 | 75.7 | 78.5 | 76.0 | - | - | - | - |
| MathBERT (**s=3**) | 84.2 | 83.9 | 69.1 | 64.5 | 63.3 | 52.2 | 66.3 | 64.0 | 67.4 | 58.7 | - | - |
| MathBERT (**s=4**) | 67.1 | 68.4 | 59.5 | 52.6 | 62.3 | 62.1 | 48.5 | 47.9 | 68.6 | 68.0 | 51.8 | 29.0 |
| SciBERT-uncased (**s=2**) | 92.5 | 92.6 | 72.9 | 70.4 | 86.8 | 86.1 | 90.0 | 90.2 | - | - | - | - |
| SciBERT-uncased (**s=3**) | 88.9 | ***89.4*** | 82.1 | ***81.9*** | 70.3 | 66.4 | 70.9 | ***72.2*** | 80.6 | ***81.8*** | - | - |
| SciBERT-uncased (**s=4**) | 76.3 | **76.5** | 69.5 | 66.8 | 68.6 | 65.9 | 60.7 | **59.6** | 76.9 | **77.9** | 59.3 | 57.4 |
| SciBERT-cased (**s=2**) | 92.6 | **93.1** | 85.3 | 87.1 | 89.8 | **90.2** | 91.0 | **91.7** | - | - | - | - |
| SciBERT-cased (**s=3**) | 77.2 | 72.4 | 72.7 | 67.2 | 61.0 | 44.1 | 50.8 | 29.5 | 52.9 | 12.8 | - | - |
| SciBERT-cased (**s=4**) | 71.0 | 70.9 | 65.1 | 64.6 | 66.6 | 65.4 | 47.0 | 42.9 | 77.9 | 74.9 | 52.7 | 11.0 |
| Encoder Average (**s=2**) | 88.6 | 89.0 | 80.7 | 79.9 | 85.6 | 85.3 | 86.5 | 86.8 | - | - | - | - |
| Encoder Average (**s=3**) | 81.5 | 80.3 | 72.9 | 69.8 | 65.7 | 57.5 | 59.2 | 53.2 | - | - | - | - |
| Encoder Average (**s=4**) | 68.0 | 70.2 | 61.5 | 61.2 | 63.3 | 64.2 | 48.4 | 50.3 | - | - | - | - |
| GPT-3.5 (**s=2**) | 66.0 | 72.6 | 65.5 | 72.5 | 59.0 | 65.3 | 53.0 | 63.3 | - | - | - | - |
| GPT-3.5 (**s=3**) | 57.0 | 64.2 | 61.5 | 67.0 | 60.5 | 65.5 | 46.0 | 54.2 | 56.5 | 64.5 | - | - |
| GPT-3.5 (**s=4**) | 51.5 | 59.1 | 49.5 | 56.3 | 54.0 | 59.6 | 44.5 | 52.8 | 56.0 | 62.7 | 59.0 | 67.7 |
| GPT-4 (**s=2**) | 88.0 | 88.5 | 87.5 | **88.2** | 82.5 | 81.1 | 64.5 | 66.4 | - | - | - | - |
| GPT-4 (**s=3**) | 77.5 | 77.4 | 77.5 | 76.7 | 78.5 | ***77.2*** | 54.5 | 55.0 | 73.5 | 77.4 | - | - |
| GPT-4 (**s=4**) | 68.0 | 68.0 | 69.0 | _69.6_ | 66.0 | 64.6 | 42.0 | 42.6 | 76.0 | 76.9 | 77.5 | ***80.2*** |
| Encoder (steps avg) | 79.4 | 79.8 | 71.7 | 70.3 | 71.5 | 69.0 | 64.7 | 63.4 | - | - | - | - |
| GPT-3.5 (steps avg) | 58.2 | 65.3 | 58.8 | 65.3 | 57.8 | 63.5 | 47.8 | 56.8 | - | - | - | - |
| GPT-4 (steps avg) | 77.8 | 78.0 | 78.0 | 78.2 | 75.7 | 74.3 | 52.2 | 54.7 | - | - | - | - |

Table 1: Model performance on the *Derivation Step Classification* task. Bold numbers denote highest F1 scores for **2-step** derivations. Bold italic numbers denote highest ***3-step*** scores. Bold, italic, and underlined numbers denote highest ***4-step*** scores.

| | Static | | VR | | EC | | Easy | |
|---|---|---|---|---|---|---|---|---|
| | Acc | F1 | Acc | F1 | Acc | F1 | Acc | F1 |
| BERT-base-uncased (**int**) | 90.0 | 90.7 | 68.8 | 70.4 | 75.1 | 78.0 | 62.7 | **72.9** |
| BERT-base-uncased (**diff**) | 75.9 | 80.3 | 64.9 | 73.3 | 62.2 | ***69.8*** | 55.1 | 69.1 |
| BERT-base-cased (**int**) | 93.0 | 93.4 | 71.6 | **77.7** | 85.2 | **86.7** | 63.8 | 71.8 |
| BERT-base-cased (**diff**) | 74.2 | 77.9 | 64.2 | 72.4 | 60.3 | 64.9 | 56.7 | ***69.6*** |
| MathBERT (**int**) | 92.2 | 92.3 | 74.4 | 75.8 | 74.4 | 71.8 | 58.6 | 68.6 |
| MathBERT (**diff**) | 84.7 | 85.9 | 59.7 | 48.1 | 58.4 | 47.3 | 56.1 | 50.0 |
| SciBERT-uncased (**int**) | 96.8 | 96.8 | 65.6 | 74.4 | 54.1 | 15.8 | 62.6 | 71.1 |
| SciBERT-uncased (**diff**) | 91.8 | 92.3 | 72.6 | ***76.5*** | 66.8 | 58.1 | 55.2 | 67.8 |
| SciBERT-cased (**int**) | 97.1 | **97.2** | 68.1 | 75.8 | 54.2 | 17.0 | 58.0 | 67.1 |
| SciBERT-cased (**diff**) | 92.3 | ***92.7*** | 70.9 | 76.5 | 65.4 | 54.6 | 61.5 | 72.3 |
| Encoder Average (**int**) | 93.8 | 93.2 | 69.7 | 74.8 | 68.6 | 53.7 | 61.1 | 70.3 |
| Encoder Average (**diff**) | 83.8 | 85.8 | 66.5 | 69.4 | 62.6 | 58.9 | 56.9 | 65.8 |
| GPT-3.5 (**int**) | 49.5 | 56.3 | 49.5 | 56.3 | 51.5 | 60.1 | 54.5 | 58.1 |
| GPT-3.5 (**diff**) | 49.0 | 55.3 | 48.5 | 54.2 | 53.0 | 65.7 | 54.5 | 59.2 |
| GPT-4 (**int**) | 64.0 | 60.0 | 67.0 | 64.1 | 66.5 | 68.5 | 57.5 | 56.4 |
| GPT-4 (**diff**) | 59.5 | 55.2 | 61.0 | 57.1 | 66.5 | 72.9 | 68.5 | 66.3 |
| Encoders (int/diff avg) | 88.8 | 89.5 | 68.1 | 72.1 | 65.6 | 56.3 | 59.0 | 68.1 |

Table 2: Model performance on the *Calculus Classification* task. Bold numbers denote highest F1 scores for **integration** derivations. Bold italic denotes highest ***differentiation*** scores.

the average static scores of the fine-tuned encoder models, and surpass them on out-of-distribution test sets. This is demonstrated by the *Derivation Step Classification* results (Tab. 1). For instance, SciBERT-cased (**s=4**) scores 11% F1 when classifying sequences with **s=2** steps. GPT-4 obtains 80% F1 in this case. Similar generalisation is observed on the **VR** (*Variable Renaming*) set, likely due to GPT-4's exposure to vast vocabularies of mathematical symbols (*e.g.,* Greek symbols), and the **EE**

(*Expression Exchange*) set, likely due to GPT-4's exposure to equations with RHS functions which lessens the impact of LHS function bias.

**GPT-4 can fail to predict mathematical coherence from in-context examples alone.** The *Calculus Classification* task includes minimalistic sequences without operation annotations.Surprisingly, while GPT4 achieves the best performance on *Derivation Step Classification*, competitive performance is not observed in *Calculus Classification* despite its lower complexity. We attribute this to the fact that, unlike BERT, GPT is not fine-tuned on a specific operation and in-context examples alone might not contain enough information to consistently discriminate whether a particular sequence involves either differentiation or integration. This is evidenced by the fact that both GPT models score *higher* on the **EC** (*Equation Conversion*) set. The EC perturbation changes nothing about the operation being performed, but *adds context* by writing (*e.g.*) differentiated expressions as equations with a LHS that includes $\frac{d}{dx}$. F1 scores in GPT models *increase by up to 12 points* in this case, while BERT-based scores *decrease by up to 80 points* (Tab. 2). Additionally, in *Derivation Step Classification*, both GPT models obtain comparatively lower scores on the **AR** (*Annotation Replacement*) set. This is because sufficient context has been provided for an operator that differs to the operator of interest. GPT only learns the format of the sequences and the expected output for the task in this case. However, static performance is maximised by designing the prompt in this manner (Tab. 4). We exclude **AR** scores when comparing GPT to BERT.

**GPT-3.5 cannot effectively classify mathematical reasoning.** GPT-3.5 scores *15 less* F1 points than the average encoder score of 80% on the static set, and is notably outperformed by BERT-based models on most test sets (particularly SciBERT). A notable exception are those that contain less steps (Tab. 1), where performance generally *increases* comparative to static scores. This contrasts with the significant corresponding performance drops observed in the BERT-based evaluation, indicating that GPT learns enough from in-context examples to generalise to derivations with less steps, and therefore has a deeper relative understanding of the underlying mathematics.

**Encoder models fail to generalise.** For *Derivation Step Classification*, models average 80% F1 over all static derivation lengths, and decreases due to perturbations average 10% (**VR**), 11% (**EE**), and 16% (**AR**). This is at most 4% above F1 majority baseline. BERT-uncased and SciBERT-cased fine-tuned on 2-step derivations are exceptions, but the 13 other encoder models are sensitive to at least one perturbation. All models tested do not generalise to *less* derivation steps, reaching as low as 11% F1. In *Calculus Classification* static scores average 90% and perturbations decrease this by 17% (**VR**) and 33% for **EC**. All fine-tuned models fail to generalise to perturbations and simpler examples, *with* 97% *F1 scores repeatedly dropping below* 17%. Despite the in-distribution performance, this indicates a reliance on superficial patterns rather than the underlying rules of the operators.

## 5.1 RELATING OPERATORS TO MODEL GENERALISABILITY VIA PAIRWISE ANALYSIS

We can alternatively measure generalisability by examining the proportion of examples where predictions involving static sequences are correct, while predictions for mathematically equivalent perturbed sequences are incorrect. Defining an example to consist of a static sequence grouped with its perturbed equivalents, if a static prediction is correct while all perturbation predictions fail, this gives a strict measure of generalisability (denoted by $G$ in Tab. 3) and complements previous analysis. These grouped examples allow examination of how well models understand each operator, and can highlight their weaknesses. We identify such weaknesses shared between GPT and BERT models and discuss clear dissimilarities in a more focused discussion in this section.

**Which operators are most difficult to learn?** Substitution is dependency-wise the most complicated operation and is not associated with a fixed token (such as addition's "+"). It requires a deeper understanding of derivation structure due to a necessary reliance on dependency relations across equations (see Fig. 1). All models interpret substitution relatively poorly (**None** column, Tab. 3). Operator usage that is easier for models to recognise (and generalise) involves integration or differentiation (**All** column, Tab 3), and these are associated with specific text spans such as "\int" or "\partial". Together, this indicates that all models struggle most when operators are *not associated with fixed text spans* or when they rely on *explicitly structured dependency relations*.

**Which operations contribute to poor generalisability?** We consider the proportion of examples where static predictions succeed while all perturbation predictions fail (column $G$, Tab. 3). For BERT models, *premise renaming* and integration/differentiation *evaluation* operations rank highly,

| | Static ($S$) | Generalisability ($G$) | None | All |
|---|---|---|---|---|
| BERT | 76.0 | 3.3 | 16.5 | 60.8 |
| | $\int_E\ R\ \int\ \partial\ \times$ | $\int_E\ R\ +\ \partial_E\ -$ | $S_L\ S_R\ +\ X^O\ \times$ | $\int\ \partial\ \times\ -\ X^O$ |
| MathBERT | 79.7 | 9.0 | 13.2 | 57.2 |
| | $\int_E\ R\ \int\ \partial\ \partial_E$ | $R\ \int_E\ X^O\ \partial_E\ \div$ | $+\ S_L\ \div\ S_R\ \cos$ | $\partial\ \int\ X^O\ +\ \div$ |
| SciBERT | **87.8** | 5.0 | **7.0** | 62.7 |
| | $R\ \int_E\ \int\ -\ \div$ | $R\ \div\ \partial_E\ +\ X^O$ | $S_L\ S_R\ +\ \cos \times$ | $\int\ \partial\ -\ +\ \partial_E$ |
| GPT-3.5 | 58.2 | 2.3 | 29.7 | 45.5 |
| | $\cos\ X^O\ \partial\ \int\ R$ | $S_L\ \int_E\ S_R\ +\ X^O$ | $-\ \int_E\ \times\ +\div$ | $\cos\ X^O\ \int\ \partial\ \partial_E$ |
| GPT-4 | 77.8 | **1.7** | 12.0 | **64.7** |
| | $\cos\ \partial\ \int\ X^O\ \int_E$ | $\cos\ \times\ \partial_E\ \div\ R$ | $S_L\ S_R\ -\ R \times$ | $\cos\ \partial\ X^O\ \int \times$ |

Table 3: **Static** ($S$) represents model accuracy with respect to unperturbed examples. **Generalisability** ($G$) represents the percentage of examples where static predictions are correct and all perturbed predictions failed (lower is better). **None** represents examples where models failed predictions in all cases, and **All** represents the opposite. Symbols correspond to the top-5 most frequent (final) operators in each unperturbed sequence, where frequency is normalized with respect to operator count in the static set. $R$ is a premise renaming operator. $\int$ and $\partial$ are integration and differentiation operators. $\int_E$ and $\partial_E$ are respective evaluation operators. $X^O$ is exponentiation, $S_L$ and $S_R$ are LHS and RHS substitutions, and arithmetic symbols have their usual meaning. This table ignores the Annotation Replacement perturbation for fairer comparison between BERT and GPT.

yet this is not mirrored by GPT. To explain this difference we plot Fig. 4(a), which displays the proportion of operators ($\tilde{N}_P$) that contribute to examples where models generalise poorly at a given rank. For example, the highest ranking operator for MathBERT has $\tilde{N}_P > 25$. From Tab. 3 this operator performs *premise renaming*, denoted by $R$. Therefore, over 1/4 of examples involving $R$ contribute to poor model generalisability. In fact for all BERT-based models, the $R$ (and less so the int/diff evaluation) operators have a higher $\tilde{N}_P$ than other operators. This effect is less prominent for the GPT models. This gives a clear indication that high ranking operators have a major impact on generalisation in BERT models, and it is likely that other factors (such as the complexity of equations) are more impactful for GPT. From Tab. 3 we can see that the highest ranking operator for GPT-4 in this context is $\cos$, which is also the highest ranking operator in examples where it generalises well. This overlap does not exist in BERT-based models and supports the conclusion that the operators themselves are not as powerful predictors of poor generalisability of GPT as they are in BERT. Fig. 4(b) accounts for each model's static and generalisation scores by multiplying $\tilde{N}_P$ by the ratio $G/S$ (Tab. 3) (and taking the negative log), resulting in a clearer separation between models and a better visualisation of generalisability rankings.

**Why is $R$ associated with generalisation failure for BERT but not for GPT?** Prior analysis points to the premise renaming operator $R$ as a useful point of comparison between fine-tuned BERT and few-shot GPT. Prompting GPT-3.5 by appending "Describe what function `renaming_premise` performs." to a static prompt (associated with GPT-3.5's generalisation failure) returns the following definition of $R$: *"the `renaming_premise` function is used to create a new expression or equation by assigning an existing expression or function to a new variable or function symbol."* This appropriate understanding persists even for perturbed prompts, and naturally extends to GPT-4. In contrast, further analysis (Appendix C) reinforces that BERT models do not share this out-of-distribution understanding. The main difference between $R$ and all other operators is that it appears in sequences *without any reference to prior equations*. The substitution operations are the opposite of this (referencing the most equations of any operators), and *both* GPT-4 and BERT frequently fail to make correct predictions given this operator. On one hand, the operator with the *least referencing* is significantly associated with generalisation failure for BERT, but not GPT-4. On the other, the operator with the *most referencing* is not significantly associated with generalisation failure in either case, as all models are not effectively learning substitution in-distribution. BERT is dependent on more localised learning where the necessary semantics is expressed within a short text span during training, rather than a span that explicitly relates to other textual elements (*e.g.,* through regular reference). In other words, *a lack of explicit discourse relations that predictably vary with the ground*

*truth obstructs models from learning latent relations that allow them to generalise.* However, the explicit relations can not be too complex (as with substitution). $R$ lends itself to generalisation failure because it lacks structured discourse relations of the appropriate complexity for BERT (that others operators do not). $R$ is simpler for GPT because of its varied exposure to structured text featuring such relations (*e.g.,* code) and obviously its relative size.

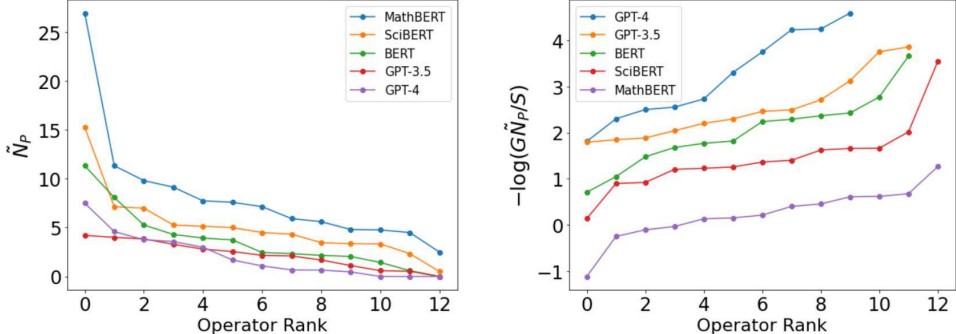

Figure 4: $\tilde{N}_P$ is the percentage of operators present in examples where models fail to generalise to perturbations. The leftmost displays how this proportion varies as a function of operator rank. The rightmost graph factors in static performance ($S$) and generalisability ($G$) scores for a clearer average ranking of the out-of-distribution performance of models.

## 6 CONCLUSION

We propose the use of math reasoning generation algorithms for developing and perturbing synthetic data for training and evaluating models. This provides a highly controllable environment within which the weaknesses of models may be examined. For example, fine-tuned BERT and few-shot GPT models struggle to identify incorrect reasoning chains when key operators explicitly rely on multiple indirect references to previous textual elements (and when they do not correspond to easy-to-identify textual markers such as "+"). This highlights the relative inability of Transformers to decode explicit structure from linearised sequences. We show that generalisation failure depends less on specific operators for GPT in comparison to BERT, and in the latter case, generalisability may be impeded by insufficient explicitly defined inter-statement relations. The inclusion of appropriate structures that relate key text spans (*e.g.,* operators) to secondary sequence elements (*e.g.,* equations) may improve the generalisability of smaller models. These models have substantial margins for out-of-distribution improvement, as we show that the application of simple perturbations can substantially affect their performance (F1 score obtained by BERT models decreases by up to 80 points).

However, smaller models may feasibly compete with GPT (in related narrowly scoped tasks) if appropriately structured inter-statement relations capturing operational semantics are incorporated during training, as the in-distribution (static) performance of the BERT models outperforms GPT-3.5 and rivals GPT-4. Few-shot GPT generalises well but *under specific conditions*. For instance, if using in-context examples as the primary mechanism for providing GPT with context (rather than relying on task descriptions), examples must contain enough information about the task *even if it is encoded in relatively complex structures*. For instance, if few-shot examples contain regularly structured dependency relations that predictably vary with respect to labels and ground truth sequences, this may aid performance comparative to examples with less structure or without explicit dependency relations. This design consideration can be useful when engineering prompts, as one might erroneously select the simplest prompt that describes the task (*e.g.,* Occam's razor), or a relatively unstructured chain-of-thought prompt (Wei et al., 2022) that minimises inter-statement dependencies.

Overall, this paper demonstrates how external symbolic engines can be leveraged to craft high-quality annotated mathematical data at scale (presently over 200K examples), which may be flexibly specialised to explore targeted weaknesses of state-of-the-art models in different settings. Future work may explore the effect of systematically increasing the number of dependency relations explicitly encoded in sequences during training or in prompts, extend the set of perturbations, or involve the fine-tuning of larger models for the purpose of improving equation derivation capabilities.

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

## A  FINE-TUNING BERT AND PROMPTING GPT

**Fine-tuning BERT.** Transformer encoders with a binary sequence classification layer are fine-tuned for 12 epochs on a 16GB Tesla V100, with a batch size of 8, and a learning rate of 5e-7, via the Transformers library (Wolf et al., 2019). We use adapted versions of the public[3] training scripts. Tokenizers pad up to a max length of 256, and the best model by F1 is selected after training. We train 25 models stemming from 5 encoders: BERT-base-uncased (Devlin et al., 2018), BERT-base-cased, SciBERT cased and uncased (Beltagy et al., 2019), and MathBERT (Shen et al., 2021). SciBERT is a version of BERT pretrained on scientific text. MathBERT is initialised on BERT-base-uncased, and pretrained on three masked language modelling tasks related to the structure of equation operator trees (Mansouri et al., 2019), and the relationship between equations and their natural language context. It delivers state-of-the-art results in formula search (Zhong et al., 2022).

**Prompting GPT.** For each task, we engineer few-shot prompts with the aim to optimise static performance with respect to the `gpt-3.5-turbo` model using the OpenAI API. The results of prompt exploration are given in Tab. 4, where the selected design is highlighted in bold. We describe this prompt below:

---

"The following examples consist of a prompt (denoted by Prompt:) and a label (denoted by Label:).

---

[3]https://huggingface.co/docs/Transformers/tasks/sequence_classification

---

`Prompt:` Sequence 1

`Label:` Label 1

`Prompt:` Sequence 2

`Label:` Label 2

`Prompt:` Sequence 3

`Label:` Label 3

`Prompt:` Sequence 4

`Label:` Label 4

Now given the following prompt, predict the label.

`Prompt:` Test Prompt"

---

The sequences all contain the same final annotation as the Test Prompt and are sampled from the training set. Additionally, an equal number of negative (Label: 0) and positive examples (Label: 1) are included as in-context examples, and these examples are shuffled. New lines are denoted by "\n". Perturbations are only applied to the Test Prompt and in-context examples are fixed to minimise examples' effect on generalisation. 200 random examples from the static test set *per subtask* (e.g. steps=2, integration) are used in the evaluation, which maps to 200 equivalent examples *per perturbation*. This totals around 4000 total examples *per GPT model*.

| Prompt Design | GPT-3.5 (F1) | GPT-4 (F1) |
|---|---|---|
| *Derivation Step Classification (steps=2)* | | |
| No task description + random examples (2 pos, 2 neg) | 61 | 83 |
| Concise task description + random examples (2 pos, 2 neg) | 50 | 83 |
| **No task description + same final operation examples (2 pos, 2 neg)** | **68** | **90** |
| No task description + same final operation examples (3 pos, 3 neg) | 68 | 87 |
| *Calculus Classification* (differentiation) | | |
| **No task description (2 pos, 2 neg)** | **55** | **55** |
| No task description (3 pos, 3 neg) | 48 | 64 |

Table 4

## B   TASK-SPECIFIC DATA FORMAT AND SIZES

| Task | Training | Validation | Static Test | Perturbed Test |
|---|---|---|---|---|
| *Derivation Step Classification* | | | | |
| 2-steps | 20K | 5K | 4K | 4K |
| 3-steps | 20K | 5K | 4K | 4K |
| 4-steps | 20K | 5K | 4K | 4K |
| *Calculus Classification* | | | | |
| integration | 32K | 8K | 4K | 4K |
| differentiation | 32K | 8K | 4K | 4K |

Table 5: The number of examples considered by models during training, validation, and evaluation.

The data generation algorithms output a derivation (Alg. 2) or expression (Alg. 1) in LaTeX and SymPy (Fig. 1). Outputs are then adapted to fit specific tasks. For the described classification tasks, a single example consists of the reasoning sequence up to the final expression or equation (Fig. 3).
**Constructing sequences.** For the *Derivation Step Classification* task, a step consists of an equation and an annotation, as described in Fig. 1 and Fig. 3. An annotation is a list comprising an operator name and its operands. Each step [an, eq] is linearised and comma separated, up to the final step. The final step annotation is separated from the derivation, and the final equation is replaced with a

negative example equation, or left unchanged.

For *Calculus Classification*, an input sequence consists of a premise expression, a variable, and the result of either differentiating or integrating. The premise expression containing *at least two* variables is initially generated, a variable is randomly selected from the premise, and the resulting expression after differentiating or integrating with respect to that variable is the ground truth. This positive example is either replaced with a negative example, or not. The three main components for each task are [SEP] separated. In the datasets for either task, this sequence is grouped with both the actual final equation and a number of negative equations. As a model encounters an example it is processed into two sequences; one including the positive equation and another including the negative. Each sequence is then paired with the corresponding classification labels. Perturbations are applied to each test set and generate an equal number of perturbed examples. The *Derivation Step Classification* datasets include 41K evaluation examples *per derivation step count*. The *Calculus Classification* datasets include 52K evaluation examples *per operation*. This equates to **227K total examples**. Tab. 5 describes the relevant sizes for the models.

**Sampling negatives.** For *Derivation Step Classification*, negatives are generated by randomly applying alternative operations (with potentially different operands) to random equations in the derivation. For *Calculus Classification*, negative examples are generated by selecting from a list of alternative premise expressions. This list includes the result of differentiating/integrating the expression with respect to other variables in the expression, and differentiating/integrating other randomly generated expressions comprised of the same symbols. The list of expressions are then ranked in terms of their Damerau-Levenshtein distance Zhao & Sahni (2019); Meadows & Freitas (2021) from the ground truth. For example, the expression $-T + sin(U)$ is differentiated with respect to $T$ to give $-1$. The corresponding negative example is $1$.

For *Calculus Classification*, the list of expressions are then ranked in terms of their Damerau-Levenshtein distance (Zhao & Sahni, 2019; Meadows & Freitas, 2021), where the closest match is the selected negative. For example, the expression $-T + sin(U)$ is differentiated with respect to $T$ to give $-1$. The corresponding negative example is $1$. The expression, variable, and candidate expression are [SEP] separated upon input to the model (*e.g.,* Fig. 3(b)).

## C    SUPPLEMENTARY MATERIAL FOR QUALITATIVE ANALYSIS

We consider (uncased) models trained on 3-step derivations. This number of steps closely reflects the average results over all step counts in Table 1. The **All** (perfect generalisation) and **Not P** (complete generalisation failure) columns of Table 6 (Appendix C) reinforce the relative generalisability gap between SciBERT and MathBERT, despite both being trained on scientific corpora, and display the top three operators by normalised frequency per generalisation category.

**Generalisation failure depends on the unpredictability of an operator.** For examples where models perfectly generalise, the operator responsible for setting up an integral (without evaluating it) is most common. This is likely because it involves prepending a unique text span "\int" to expressions either side of equations, which is easy to identify. Models generalise well to $cos$, $sin$, $exp$, and $log$ operators, likely due to their similarly predictable effect on equations associated with regular text spans. To highlight that it is likely the relative unpredictability of an operator's effect on text that leads to generalisation failure, we analyse the set of examples where *both* SciBERT and MathBERT correctly classify unperturbed sequences, but misclassify *all* perturbed sequences. Three examples are displayed in Fig. 5. The *renaming premise* operation is overwhelmingly frequent. It takes a random previously defined expression as the RHS, and defines a new function as the LHS. It does not necessarily depend on a single previous step and is non-deterministic due to random sampling of the RHS, yet it can never generate more complex equations than those previously derived (unlike other operators).

**Entailment pre-training improves generalisability.** BERT (Devlin et al., 2018) was trained on masked language modelling (MLM) and next sentence prediction (NSP) objectives. SciBERT (Beltagy et al., 2019) was further trained with scientific papers on MLM and NSP. MathBERT (Shen et al., 2021) was further trained from BERT on educational mathematical text, ranging from pre-k to graduate level difficulty. However, unlike BERT and SciBERT, MathBERT was trained to optimise performance on MLM over a large corpus. Fine-tuning generally overwrites representations learned from previous tasks (Mosbach et al., 2020), and MathBERT has likely forgotten those associated with NSP. The current classification tasks involve determining if *math context entails an expression/equation*, rather than predicting individual tokens as in language modelling. Next-equation

**(a)**

```
1
  premise
```

$$l(G) = \sin(G)$$

```
2
  ['differentiate', 1, G]
```

$$\frac{d}{dG} l(G) = \frac{d}{dG} \sin(G)$$

```
3
  renaming_premise
```

$$N(G) = \frac{d}{dG} l(G)$$

**(b)**

```
1
  premise
```

$$D(u) = \log(u)$$

```
2
  ['minus', 1, D(u)]
```

$$0 = -D(u) + \log(u)$$

```
3
  renaming_premise
```

$$a(u) = -D(u) + \log(u)$$

**(c)**

```
1
  premise
```

$$b(Q) = \cos(\cos(Q))$$

```
2
  ['divide', 1, b(Q)]
```

$$1 = \frac{\cos(\cos(Q))}{b(Q)}$$

```
3
  renaming_premise
```

$$p(Q) = \cos(Q)$$

Figure 5: Three examples of the total 15 where both SciBERT and MathBERT correctly classify *unperturbed* examples (as shown), but incorrectly classify all perturbed examples.

prediction shares greater similarity with NSP than MLM, and we therefore attribute generalisability failures of MathBERT in this context to insufficient entailment pre-training. It has struggled with entailment before relative to other BERT models (Meadows et al., 2022).

**Advantages of pre-training on structured scientific text.** SciBERT differs from the other encoders due to a distinct focus on scientific papers written in LaTeX. This offers two benefits: **(1)** Mathematical elements seen by models are written in LaTeX, so exposure to LaTeX (during both MLM *and* NSP) provides natural advantage; **(2)** Scientific papers tend to be concise and logically structured. Text spans are carefully chained to reach conclusions, so exposure to papers during training may better teach models the concept of entailment and aid performance in related tasks.

| | Static $\pm$ | All | Not P |
|---|---|---|---|
| BERT | 62.3 | 7.4 | 5.3 |
| | $R \;\; \int_E \;\; \partial_E$ | $\partial_E \;\; \int \;\; -$ | $S_L \;\; \int_E \;\; R$ |
| SciBERT | 79.6 | 21.3 | 1.6 |
| | $R \;\; \int_E \;\; \partial_E$ | $\int \;\; \partial_E \;\; \cos$ | $R \;\; X^O \;\; \times$ |
| MathBERT | 70.3 | 7.8 | 9.3 |
| | $R \;\; \int_E \;\; \int$ | $\int \;\; \cos \;\; \sin$ | $R \;\; \partial_E \;\; \int_E$ |

Table 6: **Static $\pm$** is the rate at which positive and associated negative *unperturbed* sequences are *both* correctly classified. **All** (perfect generalisation) is the percentage of examples where the static and perturbed (positive and negative) sequences are correctly classified. **Not P** (complete failure to generalise) is percentage of examples where only the static positive sequences are classified correctly, while all perturbed positive sequences are incorrect. Symbols correspond to the top three most frequent (final) operators in each unperturbed sequence, where frequency is normalized with respect to operator frequency in the static set. $R$ is a premise renaming operator. $\int$ and $\partial$ are integration and differentiation operators. $\int_E$ and $\partial_E$ are respective evaluation operators. $X^O$ is exponentiation, $\times$ is multiplication, $-$ is subtraction, and $S_L$ is LHS substitution.

## D  ALGORITHM FOR PREMISE GENERATION

Algorithm 1 The provided algorithm titled *"Generate Premise Equation"* aims to create a mathematical equation from a defined vocabulary of letters and operators. Specifically, the algorithm's process can be summarized as follows:

1. **Initialization:** Symbols and mathematical operations are defined:
   - $\mathcal{S}$ represents all symbols from the vocabulary $\mathcal{V}$.
   - $\mathcal{R}_1$ comprises unary operations like Cosine, Sine, Exponential, and Logarithm.
   - $\mathcal{R}_2$ contains binary operations such as Addition, Subtraction, Multiplication, etc.

2. **Base RHS Construction:** Depending on a randomly chosen arity (either 1 for unary or 2 for binary):
   - If arity = 1, the RHS is built by applying a random unary operation on a random symbol.
   - If arity = 2, the RHS is constructed using a binary operation on two distinct random symbols.

3. **Complexifying the RHS:** A random complexity value is selected from 0 to $\mathcal{C} - 1$. For each iteration up to the chosen complexity, the RHS's complexity is increased by applying either a unary operation on the current RHS or a binary operation between the current RHS and another random symbol.

4. **LHS Construction:** The LHS is then formulated as a function of the free symbols present in the RHS.

5. **Equation Formation:** Lastly, an equation, termed *premise*, is crafted using the finalized LHS and RHS.

In essence, this algorithm dynamically produces a mathematical equation whose intricacy varies depending on the randomly chosen operations and the selected complexity.

# E ALGORITHM FOR DERIVATION GENERATION

Algorithm 2 relies on Algorithm 1 in order to derive subsequent equations. It relies on two other procedures other than Step. The EquationDistribution function relies on the hyperparameter $p_h$, which controls the frequency that recent equations are sampled as a cubic function of $p_h$. The ExtractDerivation function is responsible for collecting all related steps from the initial longer derivation, such that a final self-contained derivation is obtained. This derivation must match the desired length, $L_f$.

**Hyperparameters.** We rely on other hyperparameters to control **1.** the selection bias towards operations being applied to more recent equations, **2.** the bias towards operators of a particular arity, and **3.** bias towards other operator subcategories.
Considering **1.**, in the 2-arity two annotation format ['operator', operand 1, operand 2], operand 1 is always an equation index. This is also true for 1-arity, and 0-arity does not require an operand. An equation is randomly sampled from a non-repeating set of derived equations. The *history hyperparameter*, $p_h$, clones an equation in the list through a cubic function of its step-wise chronological position as described above. With our default settings, the last equation in a list of three is twice as likely to be selected as input than the first. This emulates mathematicians working with recent equations, but having to occasionally sample from distant results.
Other hyperparameters work similarly, by repeating elements of lists. Considering **2.**, we bias towards 2-arity, as those contain calculus, and considering **3.** we bias towards substitution operations, differentiation, and integration. The exact form of the algorithm used to generate data for this paper is available in the linked repository on the first page.

In more formal detail, the mechanics of Algorithm 2 are as follows:

1. **Procedure Step**: This subroutine generates a single step in the derivation.
   - Sets of equations, operations, and other relevant elements are initialized from the dataset $\mathcal{D}$.
   - Based on probability parameters, the arity of the operation (either 0, 1, or 2) for this step is determined.
   - Depending on the chosen arity:
     - Arity 0: The equation and annotation for this step are directly chosen from the set $\mathcal{R}_0$.

**Algorithm 1 Generate Premise Equation**

Assumes a global vocabulary of letters, $\mathcal{V}$ and operators *e.g.,* cos, sin, etc. Accepts a complexity hyperparameter $\mathcal{C}$ that determines the maximum tree depth of the premise RHS.

```
 1: procedure PREMISE(𝒞)
 2:     𝒮 ← [Symbol(v) for v in 𝒱]
 3:     ℛ₁ ← [Cos, Sin, Exp, Log]
 4:     ℛ₂ ← [Add, Minus, Times, Power, Divide, Differentiate, Integrate]
 5:     arity ← random.choice([1,2])
 6:     if arity = 1 then
 7:         R ← random.choice(ℛ₁)
 8:         S ← random.choice(𝒮)
 9:         RHS ← R(S)
10:         LHS ← random.choice([s for s in 𝒮 if s ≠ S])
11:     else if arity = 2 then
12:         R ← random.choice([r for r in ℛ₂ if r not in [Differentiate, Integrate]])
13:         S₁ ← random.choice(𝒮)
14:         S₂ ← random.choice([s for s in 𝒮 if s ≠ S₁])
15:         RHS ← R(S₁, S₂)
16:         LHS ← random.choice([s for s in 𝒮 if s not in [S₁, S₂]])
17:     end if
18:     complexity ← random.choice(range(𝒞))
19:     for i ∈ range(complexity) do
20:         arity ← random.choice([1,2])
21:         if arity = 1 then
22:             R ← random.choice(ℛ₁)
23:             RHS ← R(RHS)
24:         else if arity = 2 then
25:             R ← random.choice(ℛ₂)
26:             S ← random.choice(𝒮)
27:             RHS ← R(RHS, S)
28:         end if
29:     end for
30:     LHS ← Function(LHS)(*tuple(RHS.free_symbols))
31:     premise ← Eq(LHS, RHS)
32:     return premise
33: end procedure
```

- Arity 1: An operation from $\mathcal{R}_1$ and an equation from the dataset are chosen to form the new equation.
- Arity 2: An operation from $\mathcal{R}_2$, an equation from the dataset, and another element are selected to shape the equation.
- If the formed equation is deemed valid through certain checks it is returned; otherwise, None is returned.

2. **Main Derivation Loop**: This section assembles the derivation.

- The initial step of the derivation is generated using Algorithm 1.
- A pre-defined target length $L_i$ describes approximately the number of times the *Step* procedure is invoked to add new steps.
- The full derivation is extracted from the accumulated steps.
- The loop breaks when the derivation reaches a desired length $L_f$, where $L_f \geq L_i$.

To summarize, the algorithm iteratively constructs a derivation of mathematical equations, where each step is shaped by a series of operations determined by specific probabilities and conditions. It is given on the following page.

---

**Algorithm 2 Generate Equational Reasoning**

---

1: **procedure** STEP($\mathcal{D}, p_0, p_1, p_2, p_h, p_r, p_e, p_c, p_s$)
2:     $D \leftarrow [i[0]$ for $i$ in $\mathcal{D}]$
3:     $A \leftarrow [i[1]$ for $i$ in $\mathcal{D}]$
4:     $\mathcal{R}_0 \leftarrow$ [Premise] + [RenamingPremise]$\times p_r$
5:     $\mathcal{R}_1 \leftarrow$ [Cos, Sin, Exp, Log, Expand] + [EvaluateDerivatives, EvaluateIntegrals]$\times p_e$
6:     $\mathcal{R}_2 \leftarrow$ [Add, Minus, Times, Divide, Power] + [Differentiate, Integrate]$\times p_c$
           + [SubsLHSForRHS, SubsRHSForLHS]$\times p_s$
7:     elements $\leftarrow$ numbers, variables, and subexpressions from $D$
8:     arity $\leftarrow$ random.choice([0]$\times p_0$ + [1]$\times p_1$ + [2]$\times p_2$)
9:     **if** arity $= 0$ **then**
10:         $R \leftarrow$ random.choice($\mathcal{R}_0$)
11:         equation $\leftarrow R$
12:         annotation $\leftarrow R.$__name__
13:     **else if** arity $= 1$ **then**
14:         $R \leftarrow$ random.choice($\mathcal{R}_1$)
15:         $e_1 \leftarrow$ random.choice(EquationDistribution($D, p_h$))
16:         equation $\leftarrow R(e_1)$
17:         $n \leftarrow D.$index($e_1$)
18:         annotation $\leftarrow [R.$__name__$, n + 1]$
19:     **else if** arity $= 2$ **then**
20:         $R \leftarrow$ random.choice($\mathcal{R}_2$)                $\triangleright$ $R$ depends on the length of $D$
21:         $e_1 \leftarrow$ random.choice(EquationDistribution($D, p_h$))
22:         $e_2 \leftarrow$ random.choice(elements)          $\triangleright$ $e_2$ will vary depending on $R$
23:         equation $\leftarrow R(e_1, e_2)$
24:         $n \leftarrow D.$index($e_1$)
25:         annotation $\leftarrow [R.$__name__$, n + 1, e_2]$
26:     **end if**
27:     **if** equation is valid **then**            $\triangleright$ validity depends on various checks
28:         **return** equation
29:     **else**
30:         **return** None
31:     **end if**
32: **end procedure**
33: **while** True **do**
34:     $\mathcal{D} \leftarrow$ [(Premise($\mathcal{C}$), "premise")]        $\triangleright$ generate first step using Algorithm 1
35:     **while** len($\mathcal{D}$) $< L_i$ **do**           $\triangleright$ $L_i$ is an initial length of the derivation
36:         step $\leftarrow$ Step($\mathcal{D}, p_0, p_1, p_2, p_h, p_r, p_e, p_c, p_s$)
37:         **if** step is not None **then**
38:             $\mathcal{D}.$append(step)
39:         **end if**
40:     **end while**
41:     derivation $\leftarrow$ ExtractDerivation($\mathcal{D}$)
42:     **if** len(derivation) $= L_f$ **then**      $\triangleright$ $L_f \geq L_i$ is the desired length of the derivation
43:         **break**
44:     **end if**
45: **end while**
46: $\mathcal{D} = derivation$

---

