# OpenReview forum: "A Symbolic Framework for Evaluating Mathematical Reasoning with Transformers"
_ICLR.cc/2024/Conference — ICLR 2024 Conference Withdrawn Submission_

### Official Review · Reviewer_RYtb · 2023-10-30

**Soundness:** 4 excellent
**Presentation:** 3 good
**Contribution:** 3 good
**Rating:** 6
**Confidence:** 3

**Summary:**

This paper proposes a framework for generating synthetic mathematical derivations. They also propose to evaluate the generalization of Transformer-based models by applying systematic perturbations to the data. Specifically, the framework contains premise generation, derivation generation, and perturbation, based on pre-defined symbol vocabulary and hand-craft rules. The generated data is then used for a derivation step classification task and a calculus classification task. The experiments are conducted on multiple pre-trained Transformer models and the emerging large language models.

**Strengths:**

The generated datasets and the proposed tasks are challenging to current large language models. The multiple simple perturbations provide an effective evaluation of the generalization of models. This paper also proposes multiple perturbation strategies. Comprehensive experiments are conducted on pre-trained transformer-based models and large language models. The findings of comparing both types of models are inspiring. Also, the finding that the models are sensitive to simple perturbations is instructive.

**Weaknesses:**

Please see the questions listed below.

**Questions:**

Q1: It would be nice if the authors could showcase some data instances.

Q2: What does “premise (1)” in Section 3.1 refer to?

---

> ### Author Response · Authors · 2023-11-21
>
> Many thanks for the effort you put into reading this. We'll make the suggested changes in future versions!

---

### Official Review · Reviewer_o3bz · 2023-10-31

**Soundness:** 3 good
**Presentation:** 2 fair
**Contribution:** 2 fair
**Rating:** 3
**Confidence:** 3

**Summary:**

The paper proposes a synthetic way to generate mathematical derivations using a set of pre-defined rules. The authors design two classification task with the multi-step generated data. A variety of finetuned BERT-based models and GPT-3.5 and GPT4 are evaluated on the benchmark.

**Strengths:**

1. The paper proposes an environment to automatically generate and evaluate mathematical expressions to evaluate large language models.

2. The evaluation is comprehensive with comparisons across several models and task variants.

**Weaknesses:**

1. One major concern I have is what the benefit is to develop such synthetic framework for large language model training and evaluation. The task format is synthetic and the derivation steps have no particular meaning and ingenuity associated with them. One can indeed evaluate the performance of various models on this benchmark. But does it tell us how good these models will be on real-world dataset.

2. There has been various related effort in creating synthetic framework that generates mathematical expression or code expression for evaluation. The novelty is thus limited.

**Questions:**

1. Can the authors explain what is the novelty of their proposed framework compared to [1, 2, 3] which also create synthetic benchmark datasets?

2. Can the authors show any correlation between performing well on synthetic benchmark and real-world datasets?

3. Would the data generated from the synthetic framework serve as additional pretraining data that can be beneficial to various models? I am trying to understand what the possible benefits we could have by developing such framework.

4. What does negative example mean in page 4 and 5?

[1] Wu, Yuhuai, Albert Qiaochu Jiang, Jimmy Ba, and Roger Grosse. "Int: An inequality benchmark for evaluating generalization in theorem proving." arXiv preprint arXiv:2007.02924 (2020).

[2] Li, Wenda, Lei Yu, Yuhuai Wu, and Lawrence C. Paulson. "Isarstep: a benchmark for high-level mathematical reasoning." arXiv preprint arXiv:2006.09265  (2020).

[3] Devlin, Jacob, Jonathan Uesato, Surya Bhupatiraju, Rishabh Singh, Abdel-rahman Mohamed, and Pushmeet Kohli. "Robustfill: Neural program learning under noisy i/o." In International conference on machine learning, pp. 990-998. PMLR, 2017.

---

> ### Author Response · Authors · 2023-11-21
>
> We appreciate the feedback. We'll make things clearer in future versions.

---

### Official Review · Reviewer_Y9EH · 2023-11-01

**Soundness:** 2 fair
**Presentation:** 1 poor
**Contribution:** 1 poor
**Rating:** 3
**Confidence:** 3

**Summary:**

This paper proposes to generate synthetic training data for language models using a Computer Algebra System (CAS). The authors propose a sampling procedure to generate step-by-step derivations, a series of perturbations (data augmentations), and a collection of tasks derived from the derivations. Experiments are conducted with GPT-3.5-turbo, GPT-4 and fine-tuned BERT models. The results show comparable in-distribution accuracy of GPT-4 and fine-tuned models, although BERT is much more sensitive to perturbations, and thus more brittle.

**Strengths:**

The paper builds on a solid intuition that CAS encode a lot of domain knowledge in the form of mathematical procedures, and these can be highly useful for both training and evaluating language models. Synthetic data is of growing interest for researchers training LLMs, with recent works (tinystories, phi-1.5, tinyGSM) showing small models achieving impressive results when trained on high-quality synthetic data. Currently, CAS to generate training data is underexplored as far as I'm aware, and I have been following this literature.

This framework should scale well without requiring a strong teacher model to generate data (like tinystories et al, which use GPT-4). Moreover, the solutions are much more likely to be correct than when sampling from a teacher LLM, which is much more prone to errors than an established CAS.

**Weaknesses:**

In its current form, the paper is quite hard to understand. The core technical bits are confusing (perhaps too much delegated to the Appendix). For example, I can't tell much about how the sampling procedure works. This is very important, because it's highly non-trivial to sample symbolic problems with reasonable complexity and diversity. If I understand, what is sampled is always an equation, so of the form L = R. Then, L and R are sampled from a grammar with 18 operators, some unary, some binary, some terms with arity 0 (variables? constants?). These operators include 'integral' and 'derivative', and trigonometric functions, so the problems span algebra, calculus and trigonometry. If this is the case, it seems extremely unlikely to generate valid problems naïvely, and right now 3.2 does not clarify this, or how often the procedure works and fails, or how many of the problems end up being of each kind (algebra, calculus, a mix, etc). Without these, I have little intuition for how the data looks like.

I'm also confused by what the tasks actually are. The paper describes two tasks as "classification", but if I understand them, they are not about classification in the classical sense (of labeling examples with a given finite set of labels). Perhaps the questions generated are multiple-choice. If that's the case, I'm missing what is the number of options, and this determines what is chance performance, which I need to interpret the results. I was hoping the Appendix would contain actual prompts, but Appendix A only has a high-level template. In either case, classification or generation, it would be helpful to include an actual example of a task. Figure 1 supposedly has an example, but I can't tell what the model receives and what it has to predict.

Overall, my main concern is about the choice of tasks to generate and evaluate. The tasks seem a bit arbitrary, and they don't seem to assess mathematical _reasoning_ in a meaningful way. If you want GPT-4 (or 3.5) to compute an integral, you'd ask it to do it step-by-step before outputting an answer. If computing an integral is one of the tasks here (seems to be what "Calculus Classification" refers to), it seems like the task format is to predict the result in a single step, like was done in Lample & Charton, 2019, who trained a seq2seq model for this. This is incompatible with modern evaluation of LLMs in reasoning tasks, since we know direct prediction won't work very well, not even in simpler tasks like GSM8k. Now, since both GPT-4 and even SciBERT get high numbers in Table 1, the proposed tasks must be simpler than I thought, then, for direct prediction to work at all. In either case, this makes me unsure about what reasoning capability is being evaluated from the models, which is supposed to be the main contribution of the paper.

**Questions:**

- What exactly are the two "classification" tasks? It would help a lot to give concrete input/output examples for each.
- What fraction of the generations fail? How long does it take to generate one valid example?
- What is the distribution of "kinds" of problems that are generated? I'm assuming not all of them involve calculus, for instance.
- What is the simplest kind of problem you generate? Would those be solving linear equations on one variable?
- For prompting gpt-3.5 and gpt-4, did you use direct prediction, or do you ask the model to generate a chain of thought first?

---

> ### Author Response · Authors · 2023-11-21
>
> To avoid such misunderstandings we should make things clearer on our side, thank you for your comments.

---

### Official Review · Reviewer_5bKm · 2023-11-07

**Soundness:** 3 good
**Presentation:** 4 excellent
**Contribution:** 3 good
**Rating:** 6
**Confidence:** 4

**Summary:**

This paper presents a novel approach for producing synthetic mathematical derivations using a computer algebra system by perturbing various aspects of mathematical data, including syntax and semantics. They considered both SymPy and LaTex-based sequences for these tasks. The aim is to assess the effectiveness of Transformers in solving symbolic and quantitative reasoning problems while offering a comprehensive framework for constructing extensive and reliable benchmarks. They created two tasks *Derivation Step Classification* (predicting the final result of a sequence of operations) and *Calculus Classification* (predicting which operation is used, differentiation or integration on sequence with a single inference step). They evaluated BERT-based models, GPT-3.5&4, and the encoder model on these tasks.

*Takeaways*

* Application of simple perturbations can substantially affect the performance of small models such as BERT compared to GPT
* If operational semantics are adequately included in the training process, BERT-based models performance improves.
* Even for GPT, examples must contain enough information about the task in the context for a better generalization.

**Strengths:**

* A framework/method to create high-quality mathematical data
* A detailed analysis of the effect of operators on task performance, generalizability
* This in detailed study led to observations on the performance of models like BERT and GPT and ideas on how to improve them.

**Weaknesses:**

* Many specialized models are created for solving math equations. None of these models are used in the benchmark.

[1] Noorbakhsh, Kimia, et al. "Pretrained Language Models are Symbolic Mathematics Solvers too!.

* Prompts used are very simple. Better analysis can be done from the chain of thought kind of prompting.

**Questions:**

NA

---

> ### Author Response · Authors · 2023-11-21
>
> Thank you for the comments! We'll consider them in the revision of the paper.